# Willingness and perceived ability to pay for Uganda's proposed national health insurance scheme among informal sector workers in Iganga and Mayuge districts, Uganda: A contingent valuation method

Noel Namuhani[1]*, Angela N. Kisakye[1,2], Suzanne N. Kiwanuka[1]

**1** Department of Health Policy Planning and Management, Makerere University School of Public Health, Kampala, Uganda, **2** African Field Epidemiology Network, Kampala, Uganda

\* nnamuhani@musph.ac.ug

## Abstract

High out-of-pocket expenditures (OOPE) continue to make access to healthcare a challenge among informal sector workers in most low-income countries. In response, Uganda has proposed a national health insurance scheme (NHIS), although the informal sector's willingness and ability to pay for it remains unexplored. This study assessed the willingness and perceived ability to pay for the proposed NHIS, along with their determinants, among informal sector workers in Iganga and Mayuge Districts, Uganda. A cross-sectional study was conducted in Iganga and Mayuge Districts between April and May 2019. A contingent valuation method using the bidding game technique was used to elicit the willingness to pay (WTP). A total of 853 informal sector workers were randomly selected and all interviewed (100% response rate). Seven focus group discussions (FGDs) were also conducted. Logistic regression analysis was done to identify the determinants of WTP for the proposed NHIS. Qualitative data were analyzed thematically. The majority 695/853, (81.5%) and 633/853, (74.2%) of the respondents were willing and believed that they would be able to pay for NHIS, respectively. The median WTP was UGX 25,000 (USD 6.8) annually. WTP was significantly associated with being a fisherfolk (AOR: 1.70 95%CI: 1.04-2.79, P = 0.035), being in the fourth wealth quintile (AOR: 2.98, 95% CI: 1.56–5.65), "not having heard" about health insurance (AOR: 0.50 95%CI: 0.23-0.86, P = 0.032), and "not having membership in a savings group" (AOR: 0.51, 95%CI: 0.34-0.76, P < 0.001). Most FGD participants reported willingness to pay for the proposed scheme; however, some doubted their ability to do so due to extreme poverty levels and unstable income. The WTP for the proposed NHIS in the informal sector was high. However, raising awareness and considerations for the indigent when setting appropriate premiums should be a priority.

**Data availability statement:** The dataset analyzed for this study has been uploaded as a supplementary file.

**Funding:** The authors received no specific funding for this work.

**Competing interests:** The authors declare that they have no competing interests.

## Introduction

Globally, there is a growing commitment to achieving Universal Health Coverage (UHC), which is recognized as a key component in attaining the Sustainable Development Goals (SDGs) [1]. UHC seeks to improve access to quality and affordable health services due to high out-of-pocket expenditures (OOPE), a major concern in many African countries [2]. In 2021, it was estimated that about 4.5 billion people (more than half) in the world were not fully covered by essential health services, while about 2 billion (a quarter) of the world's population were facing financial hardship due to out-of-pocket health spending [1]. This placed about 150 million people at risk of catastrophic healthcare costs annually, pushing over 100 million people into poverty worldwide [2]. According to the World Bank, direct payments for healthcare services account for 40% of health care expenditure in developing countries [3]. Yet, these countries bear a disproportionate burden of preventable disease, including the rising prevalence of non-communicable diseases [4]. Therefore, the need for health financing reforms to reduce OOPE, address the disease burden and inequities cannot be overstated.

In Uganda, although the national health accounts reported a reduction in the OOPE on health as a share of current health expenditure (CHE) from 38.6% in the financial year 2018/2019 to 27.4% in the financial year 2020/2021, the OOPE remains significantly high [5]. The median monthly household out of pocket payment for health care services in 2023/2024 was 32,000 Uganda shillings (USD 8.9) [5]. The catastrophic health expenditures remain high with 28.7% of the households spending more than 40% of their non-food expenditure on health care [6]. This challenge is compounded by the complex health financing landscape, where the government health expenditure per capita allocation is very low, estimated at USD 11.6 [7]. The health sector also relies heavily on donor funding, which accounts for 50.3% of the total health expenditure, although this has significantly reduced over the years [7]. Although Uganda abolished user fees in public health facilities more than two decades ago, accessing healthcare is still associated with several expenses, including transport, drugs, diagnostics, and consultancy fees, among other expenses related to personnel and capital costs [8].

More than 60% of the global population are informally employed but in Uganda, this amounts to over 80% of the labor force [9]. This includes peasant farmers, commercial cyclists, fishermen, and those employed in unregistered or small-scale enterprises. In Eastern Uganda, it is estimated that 59% of the people in the informal sector cannot afford to access quality health care due to high poverty levels and OOPE for health care [10].

The implementation of health insurance schemes is one of the main strategies to address OOPE on health care in low and middle-income countries [2,11]. Health insurance has the potential to reduce the financial barriers to accessing healthcare and to protect individuals and families against the risk of unpredictable health care expenditures [12].

Consequently, high coverage and enrolment in the proposed NHIS are essential for the success and sustainability of the scheme, since premiums are the main

proposed source of funding. Unfortunately, enrolment in health insurance schemes remains low across most African countries, ranging from 2% to 40% [13–15]. The poor enrolment rates in various countries have been attributed to several factors including low incomes, large household sizes, long distances to health facilities, limited awareness, poor quality of health care services, inappropriate benefit packages, lack of trust in the systems, and high illiteracy levels. These factors are amplified in the informal sector [16–18]. Informal sector workers are often of low socioeconomic status compared to the formal sector workers, which limits their ability to pay for health insurance [18]. Moreover, the informal sector is poorly organized and characterized by unpredictable income streams, making it difficult to enroll and register workers into NHIS and to collect regular contributions from them [15,19].

Uganda is in the process of developing a NHIS to address the high OOPE compromising access to services. However, given the high poverty levels among Ugandans and more so in the informal sector, the policy question of WTP and ability to pay remains unanswered. Eastern Uganda is the poorest region in Uganda, with 35.7% of most households living below the poverty line [9]. The WTP for the informal sector, which makes up 80% of Uganda's population, is critical to the attainment of equity in access to care and the core risk pooling function of a health insurance scheme. High WTP would translate into high enrollment rates, which is more likely to foster the financial sustainability of the scheme [12].

The NHIS Bill, (2019) proposes mandatory enrollment for all Ugandans, covering both formal and informal sector workers. Each person will be required to pay a premium, the amount of which will be determined by the board, to access the scheme. The main sources of funding for the proposed scheme are premium contributions and government contributions for the indigent. Services will be provided in both public and accredited private health facilities. In 2021 the bill was passed by Parliament but was not assented to by the president due to disagreements stemming from inadequate stakeholder consultation. To date (2026), the bill is still being discussed by the MoH and other stakeholders pending re-tabling to the Parliament of Uganda.

Overall, the willingness to pay for community-based health insurance and private health insurance in Uganda has been reported to be high (77.9%) [20,21]. But there is a paucity of information on the willingness and ability to pay for the proposed NHIS scheme, particularly within the informal sector. Understanding WTP and its associated factors is crucial in designing strategies to improve enrollment rates in the scheme. Therefore, this study aimed to assess WTP and the perceived ability to pay for the proposed NHIS, along with the factors influencing WTP among informal sector workers. The study's findings will be used by the Ministry of Health (MOH) to determine affordable premiums and design strategies to ensure maximum participation, especially among informal sector workers.

## Materials and methods

### Study design

This cross-sectional study employed a convergent parallel mixed methods design. Quantitative and qualitative data were collected concurrently. The study was conducted from 10/04/2019–23/05/2019. This study adopted a contingent valuation method (CVM) using the bidding game technique to directly elicit or measure willingness to pay for a hypothetical (not yet on the market) health insurance package [22]. This approach was chosen because it has been demonstrated to be an effective way of assessing WTP with minimal bias for goods that are not presently available on the market. CVM creates a hypothetical marketplace in which no actual transactions are made; hence, this approach has been successfully used for commodities that are not exchanged in regular markets. The CVM approach assumes that people have had no prior experience buying the health service since it is yet to be put on the market. Instead, it solicits people's WTP based on their expectations and the anticipated benefits they attach to the service [23]. In the absence of markets for public goods, this method presents consumers with hypothetical markets in which they can buy a health insurance policy. Since the elicited WTP values are contingent upon the particular hypothetical market described to the respondent, this approach is called the contingent valuation method [24].

## Study setting

The study was conducted in the Iganga and Mayuge Districts of Eastern Uganda. These rural districts were selected purposively because 80% of the Ugandan population lives in rural areas and these are among the poorest districts in the country. They also have 80–85% of the population belonging to the informal sector [9]. Iganga District is bordered by Kaliro District to the north, Bugweri District to the east, Mayuge District to the south, Jinja District to the southwest, and Luuka District to the west. With a total population of 504,197 people, 9.9% of the households are 5 km or more from the nearest health facility, and 19.9% are near a public health facility [25]. It is composed of 3 counties, 8 sub-counties, 66 parishes, and 395 villages. The major economic activity is farming. Mayuge district is in south-eastern Uganda with a population of 473,239 and a growth rate of 3.5%, and it has thirteen sub-counties. The major economic activities are farming and fishing. Mayuge district is bordered by Lake Victoria to the south, as well as Jinja and Bugiri districts. It has a total of 33 landing sites which support the fishing activities of the community. Uganda's healthcare system is pluralistic with both public and private health providers.

## Study population

The study targeted the four major categories of informal sector workers: 1) farmers; 2) fisherfolk; 3) commercial cyclists; and 4) the business community (traders and market vendors) in the study area. The informal sector was chosen because it constitutes the largest (80%) portion of the population, without which the sustainability of the scheme would be questionable.

## Sample size

The sample size was calculated using the Kish formula [26], considering a 95% confidence interval, a 50% expected level of WTP, a 5% level of precision, a design effect of 2, and a 10% non-response rate. This yielded a total sample size of 853 informal sector workers. In case a sampled household or respondent were not available, they were replaced until the total sample of 853 was achieved. The sample size for each district was calculated proportionate to size (Table 1).

Four focus group discussions were conducted with women (four FGDs), and three FGDs with men (in the informal sector). This number depended on the level of saturation of information and was determined by doing preliminary analysis while still in the field.

## Sampling procedure

Stratified random sampling was used in this study. In Iganga district, the informal sector workers were divided into three major categories: farmers, commercial cyclists, traders/market vendors. In Mayuge district, the informal sector workers were divided into four categories: fishermen, farmers, commercial cyclists, and market vendors. A random sample from each category was taken based on a number proportional to the category's size when compared to the population. In Iganga district, two sub-counties were randomly selected, and five villages from each sub-county were also randomly selected. A proportionate sample of 286 households from the 10 villages was selected, and the household heads were systematically selected for interviews. Of the sampled households, 17 were unavailable at the time of data collection. In these cases, an eligible replacement household was systematically selected from the same sampling unit, until the

**Table 1. District sample sizes.**

|  | Iganga | Mayuge | Total |
|---|---|---|---|
| Population N (%) | 504,197 (51.6) | 473,239 (48.4) | 97,7436 (100.0) |
| Sample population | 0.516*853=**440** | 0.484*853=**413** | **853** |

required sample size was achieved. Seven commercial cyclist stages located along main roads and junctions were randomly selected, and 10 cyclists available at the time of the study were randomly selected to participate in the interviews. A total of 84 traders were selected systematically from the two main markets for interviews. In Mayuge District, a total of 10 out of 33 landing sites were randomly selected, and 248 fishermen were selected from the 10 landing sites proportionate to size. Sixty-two traders were randomly selected from three main markets systematically, and 62 commercial cyclists were randomly selected from six stages located along main roads in Mayuge town (Table 2). Men and women from each of the categories were purposively selected for the focus group discussions. These were selected and identified with the help of local leaders based on community members who could express themselves but were also residents and worked in the informal sectors of interest.

## Data collection methods and tools

The WTP was assessed using a CVM using the "bidding game" technique. The bidding game was used to determine the maximum premium that a respondent would be willing to pay for the proposed NHIS plan. A bidding game technique was chosen because of its ease of implementation and its minimal biases. In addition, many studies have found the bidding game to be very reliable [27,28]. The respondent was asked if he or she would be willing to pay the starting bid. If the respondent agreed, the interviewer would raise the bid by 10% of the first bid and again ask if the respondent would be willing to pay the new bid. The interviewer continued to present the different bids until the respondent expressed an unwillingness to pay. On the other hand, if the respondent expressed unwillingness to pay the starting bid, the interviewer lowered the bid by 10% and repeated the query. This process continued until a bid was reached where the respondent would be willing to pay. The starting bid was 20,000 UGX (5.4 USD), since this was the average premium for beneficiaries in the existing community-based health insurance schemes in Luweero, Western Uganda, and Jinja [29]. The proportion of those willing to pay, the average, and the median amount of money people would be willing to pay were computed. The perceived ability to pay was measured by asking respondents whether they would be able to pay premiums for health insurance per year and how much they could afford without foregoing other basic needs.

**Table 2. Sampling procedure per category of informal sector.**

**Iganga district (n = 440) (Iganga district planning office, 2018)**

| Category | Farmers/peasants | Commercial cyclists | Traders/vendors | Fisherfolk |
|---|---|---|---|---|
| Proportion in the district | 0.65 | 0.16 | 0.19 | |
| Sample size | (0.65*440) =286 | (0.16*440) =70 | (0.19*440) =84 | – |
| Selection procedure | 2 sub-counties (SC) and 5 villages from each SC were randomly selected. Households were selected proportionate to size. | 7 commercial cyclist stages in the town were randomly selected, cyclists at each stage randomly selected to be interviewed | Traders were systematically selected for the interviews from two main markets | – |

Mayuge district (n = 413)

| Category | Farmers/peasants | Commercial cyclists | Traders/vendors | Fisherfolk |
|---|---|---|---|---|
| Proportion in the district | 0.1 | 0.15 | 0.15 | 0.6 |
| Sample size | (0.1*413) =41 | (0.15*413) =62 | (0.15*413) =62 | (0.6*413) =248 |
| Selection procedure | One sub-county and two villages were randomly selected, the number of households were determined proportionate to size. | 6 commercial cyclist stages in town were randomly selected, later cyclists on stage at that time were randomly selected to be interviewed | Traders were systematically selected from 3 main markets | 10/33 landing sites were randomly selected. Fishermen at each site were randomly selected, proportionate to size |
| **Total samples** | **327** | **132** | **146** | **248** |

Both quantitative and qualitative data were collected concurrently (Convergent parallel mixed methods). A semi-structured questionnaire was used to collect quantitative data such as sociodemographic variables, WTP, perceived ability to pay, and associated factors from informal sector workers. The authors developed the tool based on literature and their prior knowledge on NHIS. The tool also captured a few responses qualitatively. The data collection tools were translated to the local language and then back translated to English to check whether the translated questions still retained construct validity. The principal investigator recruited and trained research assistants so that they were familiar with the statement of the problem, objectives of the study, sampling procedure, data collection tools, and plan for data collection. Research assistants were oriented to the bidding game technique for assessing WTP. They were also taught the meaning of health insurance and how it is proposed to be operationalized in the country. Research assistants were also trained in the basic techniques, such as neutrally asking questions, not inadvertently directing respondents to answers and how to record answers, especially from open-ended questions, without interpreting them. The data collection tools were pre-tested in the neighboring district, Bugweri District. This was done to test the validity and ease of application of the tools. Filled electronic forms were checked at the point of data collection for completeness, and those found incomplete were completed before the respondent was discharged. The data were cleaned and edited by the research assistants before they submitted the completed forms to the ODK server.

Focus group discussion (FGD) guides were used to collect data from informal sector workers on the determinants of willingness to pay and their perceived ability to pay for the NHIS. Each FGD was composed of 6–8 participants with a moderator and a note-taker and lasted for an average of one hour. These were conducted in the local language, and were audio recorded.

## Data management and analysis

The quantitative data were downloaded from the server and imported into STATA 14 for cleaning and analysis. A total of 853 informal sector workers were randomly selected and interviewed, achieving a 100% response rate. To achieve this target, 17 participants who were unavailable at the time of the survey were replaced. All the 853 observations were included in the analysis, and none was dropped due to errors or missing data. Data were checked thoroughly in the field to identify and address any missing information. The dataset for this study has been uploaded as a supporting file (S1 Data). Descriptive statistics, including mean, median, standard deviations, frequencies, and percentages, were obtained. The information was presented in frequency distribution tables. Principal component analysis was used to generate wealth quintiles using the demographic health survey categorization. The wealth quintiles were generated based on nine household items, which included possession of a car, motorcycle, bicycle, radio, television, mobile phone, piece of land, owning a house, and having animals. Inferential statistics were obtained using logistic regression. This was done in two stages: 1) bivariate analysis was done to determine the potential variables associated with willingness to pay for health insurance. Crude odds ratios at a 95% confidence interval were obtained to measure the potential association. 2) A multivariable analysis was done to determine the actual factors associated with WTP. All variables that were statistically significant at bivariate analysis and the key factors known in the literature to be statistically significant, such as socio-economic status, occupation, and potential confounders like residence, were included in the model. Variables were tested for multicollinearity prior to being included in the model. The model was built step-by-step while testing it using *lfit* until the best model was obtained. Variables that remained statistically significant at 95% confidence were considered the factors associated with WTP for health insurance. Adjusted odds ratios were obtained as measures of association.

Qualitative data were analyzed manually using a thematic-inductive approach. The recordings were transcribed verbatim and translated from the local language (Lusoga) to English. Transcripts were read by more than two independent analysts who identified codes, sub-themes, and then organized them into themes. The information was triangulated with the quantitative data findings to gain a deeper understanding.

### Ethics statement

The study was approved by the Makerere University, School of Public Health Higher Degrees, Research and Ethics Committee (2017/HD07/1631U). At the district level, permission to conduct the study was obtained from the local leaders and the District Health Office. Permission was also sought from sub-county chiefs and local council one chairpersons. Informed written consent was obtained from each participant at the start of the study. Confidentiality and anonymity were maintained by using codes.

## Results

### Background characteristics of the respondents

Of the 853 respondents who participated in the study, 327 (38.3%) were peasant farmers, 248 (29.1%) were fisherfolk, 146 (17.1%) were traders, and 132 (15.5%) were commercial cyclists. The mean age of the respondents was 37.0 years (SD ± 11.3). The majority 667/853 (78.2%) were males, 783/853 (91.8%) of the household heads were males, and 618/853 (72.4%) resided in rural areas. The average household size was five people, with a standard deviation of 3 people. The median monthly income was UGX 150,000 (USD 40.5). Only 158/853 (18.5%) were in the wealthiest quintile, while 349/853 (41.0%) of the households were in the poor and poorest wealth quintiles (Table 3).

### Willingness to pay for the proposed NHIS

The majority, 695/853 (81.5%) of respondents were willing to pay for the proposed NHIS. Of those willing to pay, the median WTP was UGX 25,000 (USD 6.8) while the mean WTP was UGX 28,950 (USD 7.8) per person, inclusive of four dependents per year. Only 3.8% protested willingness to contribute to the scheme, and 126/853 (14.7%) had zero WTP because they believed they were too poor to pay for national health insurance (Table 3).

Most of the FGD participants were also willing to make contributions for health insurance. However, others resented making contributions, opining that payment for health services should be the role of government as noted below:

> *"Sincerely, the government is meant to provide all the medical services. So, there is no way that you can lie to us and say that the government cannot provide those services..."* **(FGD male-fishermen-Mayuge)**

Several FGD participants were not willing to pay because of poverty, while others criticized the government's misuse of taxes collected from the taxpayer to pay huge salaries for senior government officials, rather than spending on health services, as noted in the quotes below.

> *"The fact is that poverty is the biggest hindrance. Because at times you can fall sick, yet you do not have money, and you resort to borrowing from colleagues who may not even be able to help you out because they also do not have. So generally, I disagree with paying for health insurance."* **(FGD-Business Women, Mayuge District)**

> *"I am not willing to pay because we are always charged a lot of tax to register our businesses. That is all tax. Where does that tax go? Let them reduce the salary of Members of Parliament so that part of it is utilized to buy medicine. Members of Parliament have no importance in our area."* **(FFGD-Male-fishermen-Mayuge district)**

### Willingness to pay with copayments

A copayment is a cost-sharing arrangement where one covers a portion of the costs out of pocket each time one visits the health facility. Less than half, 309/695 (44.5%) of the respondents, were willing to co-pay. The mean co-payment amount

PLOS Global Public Health

**Table 3. Background characteristics of the respondents.**

| Variable | Frequency (N = 853) | Percentage (%) |
|---|---|---|
| **Sex of respondent** | | |
| Male | 667 | 78.2 |
| Female | 186 | 21.8 |
| **Sex of the household head** | | |
| Male | 783 | 91.8 |
| Female | 70 | 8.2 |
| **Residence** | | |
| Rural | 618 | 72.4 |
| Urban | 235 | 27.6 |
| **Marital status** | | |
| Married | 694 | 81.4 |
| Single | 85 | 10.0 |
| Separated | 48 | 5.6 |
| Widowed | 26 | 3.0 |
| **Age category** | | |
| 18-25 | 124 | 14.5 |
| 26-35 | 303 | 35.5 |
| 36-45 | 239 | 28.0 |
| 46-55 | 126 | 14.8 |
| >55 | 61 | 7.2 |
| **Education level** | | |
| No formal education | 51 | 6.0 |
| Primary | 445 | 52.2 |
| Secondary | 280 | 32.8 |
| Tertiary | 77 | 9.0 |
| **Religious denominations** | | |
| Muslim | 304 | 35.6 |
| Protestant | 276 | 32.4 |
| Catholic | 194 | 22.7 |
| Born again | 53 | 6.2 |
| SDA | 26 | 3.1 |
| **Occupation** | | |
| Farmer | 327 | 38.3 |
| Fisherfolk | 248 | 29.1 |
| Business | 146 | 17.1 |
| *Commercial cyclists* | 132 | 15.5 |
| **Wealth quintiles** | | |
| Poorest (<-1.39) | 172 | 20.2 |
| Poor (-1.40 - -0.50) | 177 | 20.8 |
| Moderate (-0.51 -0.29) | 163 | 19.1 |
| Wealthy (0.30 -1.23) | 183 | 21.5 |
| Wealthiest(1.24-4.14) | 158 | 18.5 |

was UGX 2,650 (USD 0.7), with a median of UGX 2000 (USD 0.5), each time someone accessed services. The average WTP with co-payment was UGX 23,000 (USD.6.2) and the median WTP was UGX 20,000 (USD.5.4) (Table 4).

Most of the FGD participants were not in support of co-payments, as expressed by one of the FGD participants.

*"I also do not support the idea that we pay money each time we visit the health facility, especially when we are paying every year. This will make people unable to access the services, and hence the problem of access to services will not have been solved. People are already poor, and that will mean people will continue to sell their property to go to health facilities."* (FGD Male Farmer)

### Preferred mode of payment

Most (54.1%) respondents preferred to pay annually, while 21.0% preferred to pay per harvesting season. Only 13.1% expressed a preference to pay according to their monthly income, with the majority (63.1%) preferring to pay 1 percent of their monthly income, followed by those who were willing to pay 5%.

### Perceived ability to pay for the NHIS

More than half 633/853 (74.2%) of the respondents believed that they were able to pay for health insurance, with a mean of UGX 23,000 (USD.6.2) and median of UGX 20,000 (USD.5.4) per person per year.

Regarding the perceived ability to pay, most of the FGD participants noted that most of the people are poor and most likely would not be able to make periodic subscriptions, especially if the premium exceeded what people earn. This was attested to, as indicated below:

*"Sincerely speaking, some people cannot afford to pay money; for instance, you may find a very old person who is no longer earning anything, yet they also need health care."* **(FGD-commercial cyclist-Iganga district)**

Another FGD participant, a business lady, noted that while it is a good idea, it might not be uniformly applicable to everyone.

*"I support the idea. But not very much because of the issue of affordability. Not everyone can afford to pay and note that we earn differently. Because some of us play the role of mother and father in the family, there are a lot of requirements to provide."* **(FGD Businesswomen, Iganga)**

**Table 4. Willingness to pay for the proposed NHIS annually.**

| WTP | Frequency (%) | Mean | Median |
|---|---|---|---|
| Protests (WTP=0) | 32 (3.8) | 0 | 0 |
| WTP=0 (zero WTP) | 126 (14.7) | 0 | 0 |
| WTP>0 | 695(81.5) | **28,950** | **25,000** |
| Total | 853 (100.0) | 23,600 | 20,000 |
| **Co-payment scenario** | | | |
| Amount willing to co-pay | 309/695(44.5) | **2,650** | **2,000** |
| WTP with co payment | 309/695(44.5) | 23,000 | 20,000 |

**Notes:** Protests-those protested attaching value to HI (WTP=0 but it has no economic sense). WTP=0 means Zero WTP. WTP>0 that it is positive -WTP is above zero.

Furthermore, most of the participants noted that setting premiums should be based on people's income and ability to pay, as noted below:

*"Since everyone earns differently, the charges should depend on how much someone earns. I could be able to afford UGX 50,000, but my colleague cannot at all since people in the village are used to getting free medicine. When some people are asked to pay just 500shs for drugs, they will consider it unworthy because they know that drugs/medication are freely given. So, if possible, we charge according to people's income."* **(FGD: businesswomen, Iganga district)**.

### Determinants for level of WTP for the proposed NHIS

At bi-variable analysis, wealth quintiles, savings group membership, having a family member with chronic illness, use of traditional medicine, and hearing about health insurance were statistically significantly associated with willingness to pay for health insurance. The odds of being willing to pay when in the fourth wealth quintile were 2.88, compared to the odds of being willing to pay among those in the poorest category (COR: 2.88, 95% CI: 1.60–5.18, P < 0.001). Not being in the saving group reduced the odds of being willing to pay for health insurance by 42% compared to those who had saving group memberships (COR: 0.58, 95% CI: 0.33-0.69, P < 0.001) (Table 4).

After adjusting for any confounding factors, occupation, wealth quintile, saving group membership, use of traditional medicine, and having heard about health insurance remained significantly associated with WTP for the proposed NHIS.

The odds of fishermen's WTP for NHIS were 1.70 compared to those of farmers after adjusting for other factors (AOR: 1.70, 95% CI: 1.04–2.79, P = 0.035). Respondents in the "wealthy" wealth quintile were 2.98 times more likely to pay for NHIS compared to those in the poorest wealth quintile (AOR: 2.98, 95% CI: 1.56–5.65). Respondents who were not in any saving group were 0.51 times less likely to pay for the proposed NHIS compared to those with saving group membership (AOR: 0.51, 95% CI: 0.34-0.76, P < 0.001), and individuals who had never heard about health insurance were 50% less willing to pay for the proposed NHIS compared to those who had ever heard about health insurance (AOR: 0.50, 95% CI: 0.23-0.86, P = 0.032) (Table 5).

## Discussion

According to our study, most informal sector workers would be willing and believed they would be able to pay for the proposed NHIS. WTP was significantly associated with occupation, wealth, awareness, and membership in saving groups.

Most of the informal sector workers would be willing to make contributions to the proposed NHIS. This could be perceived as support for the proposed NHIS among the informal sector workers. These findings are consistent with similar studies conducted in in Nepal (71%) and in South Sudan (68%) among the informal sector [15,30]. Similarly, a systematic review and meta-analysis of WTP found that the pooled WTP for NHIS in Africa and Asia was 71.0% [31]. It is also consistent with studies conducted in Togo (92%) [32] and 93% in Sierra Leone [33]. The high level of WTP contributes to growing evidence showing the potential of informal sector workers participating in NHIS despite their nature of having unpredictable incomes and always poor [34].

The study found a median WTP of USD 6.8 and a mean WTP of USD 7.8 per person per year, which includes coverage for four dependents. This was slightly higher than the starting bid of USD 5.4 per year, which indicates that informal sector workers would be willing to contribute more than the initially hypothesized amount. However, the WTP was lower than the 5.5 USD per person per month in Iran, 6.6 USD per person per month in Namibia, and 3 USD per month in Kenya [22,30,35]. These discrepancies in WTP may reflect variations in socio-economic status and differing perceptions of health insurance value among study populations. As Uganda deliberates (as of 2026) the NHIS bill, it is important to recognize the relatively low WTP among informal sector workers compared to other contexts. This underscores the need to design affordable premium structures tailored to the financial realities of this diverse group. The informal sector, often

**Table 5. Multi variable analysis of factors associated with WTP for NHIS scheme.**

| Variable | Level of Willingness to pay | | COR (95%CI) | AOR (95%CI) |
|---|---|---|---|---|
| | No (n = 158) | Yes (n = 695) | | |
| **Sex of respondent** | | | | |
| Male | 127(80.4) | 540(77.7) | 1.0 | |
| Female | 31(19.6) | 155(22.3) | 1.18(0.76-1.81) | 1.30(0.80-2.11) |
| **Residence** | | | | |
| Rural | 110(69.6) | 508(73.1) | 1.0 | |
| Urban | 48(30.4) | 187(26.9) | 0.84(0.58-1.23) | 1.09(0.52-2.26) |
| **Occupation** | | | | |
| Farmer | 60(38.0) | 267(38.4) | 1.0 | |
| Business | 32(20.2) | 114(15.4) | 0.80(0.49-1.30) | 0.59(0.26-1.35) |
| Commercial cyclist | 25(15.8) | 107(15.4) | 0.96(0.57-1.61) | 1.04(0.50-2.18) |
| Fisherfolk | 41(26.0) | 207(29.8) | 1.13(0.73-1.76) | **1.70(1.04-2.79) *** |
| **Wealth quintiles** | | | | |
| Poorest | 43(27.2) | 129(18.6) | 1.0 | |
| Poor | 32(20.3) | 145(20.9) | 1.51(0.90-2.53) | **1.83(1.05-3.19) *** |
| Moderate | 35(22.1) | 128(18.4) | 1.22(0.73-2.03) | 1.23(0.70-2.14) |
| Wealthy | 19(12.0) | 164(23.6) | 2.88(1.60-5.18) | **2.98(1.58-5.65) *** |
| Wealthiest | 29(18.4) | 129(18.6) | 1.48(0.87-2.52) | 1.53(0.81-2.88) |
| **In saving group** | | | | |
| Yes | 49(31.0) | 337(48.5) | 1.0 | |
| No | 109(69.0) | 358(51.5) | 0.48(0.33-0.69) | **0.51(0.34-0.76) ** ** |
| **Family member with chronic illness** | | | | |
| Yes | 36(22.7) | 241(34.7) | 1.0 | |
| No | 122(77.2) | 576(67.5) | 0.56(0.37-0.83) | 0.73(0.46-1.14) |
| **Ability to meet health care** | | | | |
| Difficult | 112(70.9) | 571(82.2) | 1.0 | |
| Easy | 46(29.1) | 124(17.8) | 0.53(0.36-0.78) | 0.68(0.43-1.05) |
| **Use Traditional Medicine** | | | | |
| No | 59(38.8) | 345(51.4) | 1.0 | |
| Yes | 93(61.2) | 326(48.6) | 0.60(0.42-0.86) | **0.71(0.48-0.92) *** |
| **Ever heard about NHIS** | | | | |
| **Yes** | 9(5.7) | 83(11.9) | 1.0 | |
| **No** | 149(94.3) | 612(88.1) | 0.45(0.22-0.91) | **0.50(0.23-0.86) *** |

Note: ** *p < 0.001, *P < 0.05*

characterized by irregular and unpredictable incomes, may struggle to afford fixed, high premiums. In Ethiopia, making premium payment systems more flexible eased payments [36].

In this study, membership in local savings groups was associated with a higher likelihood of WTP for the proposed NHIS. In Asian countries, saving groups were reported to be key in initiating community-based health insurance schemes [37]. In Kenya, savings group membership played a key role in increasing demand for health insurance especially among women and the self-employed [38]. In Cameroon similarly, savings groups were used as avenues to mobilize for health insurance enrolment among informal sector workers [39]. Savings groups provide members with a secure place to save money, generate a pool of funds, and offer loans to members in times of shock and also enable the community to meet

the premiums [37]. Given their widespread presence in Uganda and community trust [40], these groups present an opportunity to be leveraged to mobilize the informal sector to save and contribute to the proposed NHIS.

Participants belonging to the "wealthiest" quintile were more likely to pay for the proposed national health insurance scheme. Similar findings have been reported in Nigeria, Ethiopia, and Ghana, where socio-economic factors such as household income and wealth status have a direct bearing on the ease or difficulty of enrolling in a given scheme, with high premiums deterring enrollment in schemes [41–43]. Further studies conducted in Sierra Leone and Iran have shown that the WTP for the HI scheme depended on the monthly income of the respondent [33,35]. In Nepal, belonging to a lower wealth quartile was associated with a lower willingness to pay for the National Health Insurance Scheme [44]. These findings underscore the fact that WTP for NHIS is closely tied to household economic status therefore efforts to improve household income levels are essential to enhancing broad participation in the proposed NHIS.

The use of traditional medicine was negatively associated with willingness to participate in the health insurance scheme. This may be due to the perceived lack of need for allopathic health services among those who predominantly rely on traditional medicine. Studies have shown that the presence and use of alternative medicine and other forms of healthcare, such as herbalists, may negatively affect participation in health insurance schemes because NHIS are typically designed to enhance access to and utilization of allopathic medical services [45]. Studies conducted in Cameroon, Burkina Faso, Nigeria, and India have reported that those who use allopathic medical services are more willing to pay for health insurance than those who use traditional medicine [46–48]. In Uganda's pluralistic health system, the Ministry of Health and other mandated regulatory bodies need to institute measures that not only regulate the alternative medicine sector but also promote the utilization of modern healthcare services through education, trust-building, and service quality improvements.

The study noted that those who had never heard about NHIS were 50% less willing to pay compared to those who had prior information about health insurance. The level of knowledge on how health insurance operates was extremely low, and statistical tests of significance could not be used in this study. This was similar to studies conducted in other Africa and Asia where the knowledge was reported to be low [31]. A systematic review and meta-analysis of factors influencing WTP for voluntary contributions to health insurance schemes in LMICs revealed that knowledge and understanding of the functioning of health insurance schemes positively influences the WTP and participation [49,50].

## Implications and recommendations for policy and practice

This study offers valuable insights for the ongoing design of Uganda's NHIS, especially regarding the premiums and covering the informal sector. While the WTP among informal sector workers is promising, the modest contribution they are willing to make suggests that affordability must be central to the scheme design. The MoH should adopt income-sensitive premiums and flexible payment schedules, aligned with seasonal income flows such as post-harvest premium payment modalities for agricultural workers. The MoH should design targeted subsidies for the poorest households to avoid exacerbating existing health inequities.

The existing local savings groups offer an opportunity to facilitate premium collection, raise awareness, and foster a sense of local ownership. The MoH, district health teams, and partners should map these groups, build their capacity, and support them as potential entry points during the eventual roll out of the NHIS.

Low awareness and understanding of health insurance in the informal sector poses a significant barrier to participation in NHIS. The MoH, in collaboration with district health teams, should invest in sustained awareness campaigns using trusted channels such as local radio, community meetings, and influential local leaders. Messaging should demystify the concept of health insurance, explain how contributions are used, and highlight the benefits of enrollment. The government should also foster trust in the population by improving the quality of care and accountability mechanisms.

Finally, further research is needed to explore, for whom and how broader social health protection mechanisms would be extended within the informal sector effectively.

## Study limitations

As with most contingent valuation studies, the elicitation technique is always subject to bias, and the assignment of the first bid or amount is also prone to bias. However, this was triangulated with open-ended questions and qualitative data. The low level of knowledge about health insurance among respondents was another limitation in attaching value to the proposed scheme. This was addressed by educating the respondents about health insurance before eliciting their willingness to pay. Additionally, modern estimation approaches for contingent valuation data such as the Vaughan Russell method [51] could be adopted, particularly when willingness to pay (WTP) is elicited using an open-ended or continuous format. Despite these limitations, the study provides valuable insights into the WTP for the proposed NHIS among informal sector workers, who are often excluded from social protection programs.

## Conclusions

This study indicated that the informal sector workers would be willing to make contributions to the proposed NHIS. However, the premium structure should be designed with careful consideration of the ability of the informal sector to pay, given the high levels of poverty and unpredictable incomes. The study also highlights that occupation, wealth status, use of traditional medicine, prior knowledge of health insurance, and trust influence WTP for NHIS among the informal sector workers. Creating awareness and fostering trust is critical to the success of the proposed NHIS.

## Supporting information

**S1 Data. Anonymized dataset used in the study.**
(XLSX)

## Acknowledgments

We thank the District Health Officers of Mayuge and Iganga districts for their cooperation, and mobilization of the respondents to participate in the study. We also extend our sincere appreciation to the research assistants and all respondents in Iganga and Mayuge for sparing their time to contribute to this piece of work.

## Author contributions

**Conceptualization:** Noel Namuhani, Angela N Kisakye, Suzanne N Kiwanuka.

**Data curation:** Noel Namuhani.

**Formal analysis:** Noel Namuhani.

**Investigation:** Noel Namuhani.

**Methodology:** Noel Namuhani.

**Project administration:** Noel Namuhani.

**Supervision:** Noel Namuhani, Angela N Kisakye, Suzanne N Kiwanuka.

**Validation:** Noel Namuhani.

**Writing – original draft:** Noel Namuhani.

**Writing – review & editing:** Noel Namuhani, Angela N Kisakye, Suzanne N Kiwanuka.

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
