## [Decision Letter · Decision Letter 0]

5 Nov 2024

PGPH-D-24-01717

Willingness and Perceived ability to pay for Uganda’s Proposed National Health Insurance Scheme among Informal Sector workers in Iganga and Mayuge districts, Uganda: A Contingent Valuation Method.

Dear Dr. Authors,

Thank you for submitting your manuscript to PLOS Global Public Health. After careful consideration, we feel that it has merit but does not fully meet PLOS Global Public Health’s publication criteria as it currently stands. Therefore, we invite you to submit a revised version of the manuscript that addresses the points raised during the review process.

We look forward to receiving your revised manuscript.

Kind regards,

Genevieve Cecilia Aryeetey, Ph.D

Academic Editor

Journal Requirements:

Additional Editor Comments (if provided):

Reviewers' comments:

Reviewer's Responses to Questions

**Comments to the Author**

1. Does this manuscript meet PLOS Global Public Health’s publication criteria?

Reviewer #1: Yes

Reviewer #2: Yes

2. Has the statistical analysis been performed appropriately and rigorously?

Reviewer #1: Yes

Reviewer #2: Yes

3. Have the authors made all data underlying the findings in their manuscript fully available (please refer to the Data Availability Statement at the start of the manuscript PDF file)?

Reviewer #1: Yes

Reviewer #2: Yes

4. Is the manuscript presented in an intelligible fashion and written in standard English?

Reviewer #1: Yes

Reviewer #2: Yes

Reviewer #1: The manuscript addresses an important aspect of health financing in Uganda. The manuscript is also very well written with sufficient transparency about the approach and its limitations. I have included a few general comments for the authors' consideration in shaping the writeup.

1. I understand that bill has not been passed yet but it will be good to have a sense of the proposed funding sources for the scheme. It is quite unclear when the paper suggest that the scheme would unsustainable without informal sector subscribers. For example in Ghana, premiums form less than 1% of the funding and therefore increasing subscription without commensurate funding is actually a threat to the scheme.

2. It will be great if possible to also have a sense of current health financing landscape particularly in the regions being studied. How are they paying for health care? What is the level of insurance penetration?

3. The variation in WTP by age would have been very interesting even though i notice it was omitted from the multivariate analysis.

4. In the discussion it will be good to keep in mind that a wholesale call for increased enrolment will also come with budget (cost) implications. At the moment it is not clear if the premiums can support the funds required for the scheme.

Thank you for putting together this interesting manuscript.

Reviewer #2: This study sought to explore the underlying factors influencing willingness and ability to pay for NHI in Uganda. The topic is relevant for the ongoing health system reforms in Uganda, particularly in the area of healthcare financing towards achieving UHC. That said, there are pertinent issues requiring authors’ attention.

Introduction

I think the entire manuscript must be revised to demonstrate its relevance internationally beyond its current narrow scope.

Define OOPE at first use. OOP is defined but not OOPE

It is not clear what problem this study seeks to address. Please elaborate on statement of the problem. Also, please what knowledge gap is it aiming to bridge? Further, linked to the problem statement, the justification for the study is weak, authors need to strengthen it.

Please include a paragraph on the structure of the Ugandan health system, indicating clearly service coverage and payment mechanisms, and levels of the service delivery structure that services under the NHI are provided, including where the private facilities stand in the scheme of things. Also, elaborate on the NHI in terms of focus, coverage, access etc.

Methods

Authors wrote on page 9 lines 102 – 104 that this was a cross-sectional study design that involved mixed methods of data collection. Please indicate the type of mixed method study and how it was operationalized.

In describing the contingent valuation method (CVM), I think it would be appropriate for authors to weave in the relevance of this study and how it fits with the model.

Authors have done a good job at describing the sampling technique used, but not how the FGD participants were selected. Please indicate the approach used?

Discussion

I suggest authors summarize the key findings in beginning paragraph, then proceed to discuss. The discussion is haphazard and failed to properly situate the findings on the reasons for the WTP for the proposed NHI, ability to pay and NHI coverage within existing body of knowledge and contextual dynamics. Please review and properly structure the whole section.

In your discussion section, you must relate this to the international literature and point out the policy

implications or implications for practice.

Please include a section on the study’s implication for policy and practice.

Conclusion

If WTP level is high, and the majority believed they could afford to pay for the health insurance, then I think there is the need for a strong conclusion and recommendations to make this study relevant in the scheme of things. In fact, there is no clear implication for policy. Please provide a clear and succinct but cogent conclusion based on your study findings – which may be inclusive of recommendations relevant for similar contextual settings.

General comments

The paper will benefit from proofreading. There are a number of sentence construction issues and, incoherent statements and paragraphs that need attention.

**Do you want your identity to be public for this peer review?** For information about this choice, including consent withdrawal, please see our Privacy Policy

Reviewer #1: No

Reviewer #2: No

---

## [Decision Letter · Decision Letter 1]

24 Mar 2025

PGPH-D-24-01717R1

Willingness and Perceived ability to pay for Uganda’s Proposed National Health Insurance Scheme among Informal Sector workers in Iganga and Mayuge districts, Uganda: A Contingent Valuation Method.

Dear Dr. Namuhani,

Thank you for submitting your manuscript to PLOS Global Public Health. After careful consideration, we feel that it has merit but does not fully meet PLOS Global Public Health’s publication criteria as it currently stands. Therefore, we invite you to submit a revised version of the manuscript that addresses the points raised during the review process.

Please pay particular attention to the comments regarding your discussion.

We look forward to receiving your revised manuscript.

Kind regards,

Daniel Parkes, PhD

Staff Editor

Journal Requirements:

Additional Editor Comments (if provided):

Reviewers' comments:

Reviewer's Responses to Questions

**Comments to the Author**

Reviewer #2: (No Response)

Reviewer #3: (No Response)

publication criteria?

Reviewer #2: Yes

Reviewer #3: Yes

3. Has the statistical analysis been performed appropriately and rigorously?

Reviewer #2: Yes

Reviewer #3: Yes

4. Have the authors made all data underlying the findings in their manuscript fully available (please refer to the Data Availability Statement at the start of the manuscript PDF file)?

Reviewer #2: No

Reviewer #3: Yes

5. Is the manuscript presented in an intelligible fashion and written in standard English?

Reviewer #2: Yes

Reviewer #3: No

Reviewer #2: The study aimed to explore the underlying factors influencing willingness and ability to pay for NHI in Uganda. The topic is relevant for health system reforms towards achieving UHC in Uganda. Authors have addressed the majority of key issues raised in my previous reviews. However, there are still a few issues requiring authors’ attention:

Although authors have tried, I still think the discussion section is not sufficiently nuanced in a manner that will situate in global body of knowledge for international relevance. Also, the statement included on implication of the study results for policy and practice woefully superficial. I suggest authors elaborate to include actionable recommendations based on the study findings, targeted at various levels of Uganda’s health systems.

In the conclusion, results is still being presented and discussed/interpreted, instead of making concluding remarks based on the interpreted results. I believe the conclusion could still be strengthened.

Reviewer #3: Avoid unnecessary capitalization in the heading. Do not use willingness to pay and WTP concurrently. Resort to the acronym after first use. OOPE was not defined on first use. Revise that.Contrast the following statement with the burden of diseases in these countries and how that creates a paradox. "According to the World Bank, direct modes of payment for54 health care services constitute 40% of health care expenditure in developing countries (4)."The information on page 10, lines 163 to 166, is repetitive.How was the semi-structured questionnaire developed? Adapted/adopted?Often, researchers have problems integrating results in mixed-method studies. What type of mixed-methods design was employed? How did that influence the integration of both quantitative and qualitative data? Table 2: The classification of the religion is confusing. Catholicism, Protestantism, etc. are not religions but denominations or variants within the Christian fraternity. Could you correct this error?Line 264-266: "gave WTP zero" is grammatically incorrect. Should be "had a zero WTP."

**Do you want your identity to be public for this peer review?** For information about this choice, including consent withdrawal, please see our Privacy Policy

Reviewer #2: No

Reviewer #3: No

---

## [Decision Letter · Decision Letter 2]

30 May 2025

PGPH-D-24-01717R2

Willingness and perceived ability to pay for Uganda’s proposed national health insurance scheme among informal sector workers in Iganga and Mayuge districts, Uganda: acontingent valuation method

Dear Mr. Noel Namuhani,

Thank you for submitting your manuscript to PLOS Global Public Health. After careful consideration, we feel that it has merit but does not fully meet PLOS Global Public Health’s publication criteria as it currently stands. Therefore, we invite you to submit a revised version of the manuscript that addresses the points raised during the review process.

We look forward to receiving your revised manuscript.

Kind regards,

Bijoya Roy, PhD

Academic Editor

Journal Requirements:

Reviewers' comments:

Reviewer's Responses to Questions

**Comments to the Author**

Reviewer #2: (No Response)

Reviewer #3: All comments have been addressed

publication criteria?

Reviewer #2: Yes

Reviewer #3: Yes

3. Has the statistical analysis been performed appropriately and rigorously?

Reviewer #2: Yes

Reviewer #3: Yes

4. Have the authors made all data underlying the findings in their manuscript fully available (please refer to the Data Availability Statement at the start of the manuscript PDF file)?

Reviewer #2: Yes

Reviewer #3: Yes

5. Is the manuscript presented in an intelligible fashion and written in standard English?

Reviewer #2: Yes

Reviewer #3: No

Reviewer #2: This is an important topic relevant to health system reforms towards achieving UHC in Uganda. However, authors have not indicated how the key issues raised in my previous review have been addressed. Responses to my comments are not contained in the authors’ letter to the Editor. My previous queries yet to be addressed and responded to are as follows:

1. Although authors have tried, I still think the discussion section is not sufficiently nuanced in a manner that will situate in global body of knowledge for international relevance. Also, the statement included on implication of the study results for policy and practice woefully superficial. I suggest authors elaborate to include actionable recommendations based on the study findings, targeted at various levels of Uganda’s health systems.

2. In the conclusion, the results is still being discussed/interpreted instead of making concluding remarks based on the interpreted results. I believe the conclusion could still be strengthened.

Reviewer #3: "acontingent valuation method" should be "a contingent valuation method"

Study limitations should be a header on it own

**Do you want your identity to be public for this peer review?** For information about this choice, including consent withdrawal, please see our Privacy Policy

Reviewer #2: No

Reviewer #3: No

---

## [Decision Letter · Decision Letter 3]

29 Aug 2025

PGPH-D-24-01717R3

Willingness and perceived ability to pay for Uganda’s proposed national health insurance scheme among informal sector workers in Iganga and Mayuge districts, Uganda: a contingent valuation method

Dear Dr. Noel Namuhani,

Thank you for submitting your manuscript to PLOS Global Public Health. After careful consideration, we feel that it has merit but does not fully meet PLOS Global Public Health’s publication criteria as it currently stands. Therefore, we invite you to submit a revised version of the manuscript that addresses the points raised during the review process.

We look forward to receiving your revised manuscript.

Kind regards,

Bijoya Roy, PhD

Academic Editor

Journal Requirements:

Additional Editor Comments (if provided):

Reviewer #2:

Reviewer #3:

Reviewers' comments:

Reviewer's Responses to Questions

**Comments to the Author**

Reviewer #2: All comments have been addressed

Reviewer #3: All comments have been addressed

publication criteria?

Reviewer #2: Yes

Reviewer #3: Partly

3. Has the statistical analysis been performed appropriately and rigorously?

Reviewer #2: Yes

Reviewer #3: Yes

4. Have the authors made all data underlying the findings in their manuscript fully available (please refer to the Data Availability Statement at the start of the manuscript PDF file)?

Reviewer #2: Yes

Reviewer #3: Yes

5. Is the manuscript presented in an intelligible fashion and written in standard English?

Reviewer #2: Yes

Reviewer #3: No

Reviewer #2: Authors have addressed all comments and provided clarity to the queries I raised in my last review. However, I recommend that authors proofread the entire paper as there are a few sentence construction issues.

Reviewer #3: Improve the writing style.

**Do you want your identity to be public for this peer review?** For information about this choice, including consent withdrawal, please see our Privacy Policy

Reviewer #2: No

Reviewer #3: No

---

## [Decision Letter · Decision Letter 4]

19 Nov 2025

PGPH-D-24-01717R4

Willingness and perceived ability to pay for Uganda’s proposed national health insurance scheme among informal sector workers in Iganga and Mayuge districts, Uganda: a contingent valuation method

Dear Dr. Namuhani,

Thank you for submitting your manuscript to PLOS Global Public Health. After careful consideration, we feel that it has merit but does not fully meet PLOS Global Public Health’s publication criteria as it currently stands. Therefore, we invite you to submit a revised version of the manuscript that addresses the points raised during the review process.

Could you address the minor comments raised by the some reviewers including reviewing the entire manuscript to ensure there are no English language errors.

We look forward to receiving your revised manuscript.

Kind regards,

Ifunanya Clara Agu

Academic Editor

Journal Requirements:

Additional Editor Comments (if provided):

Reviewers' comments:

Reviewer's Responses to Questions

**Comments to the Author**

Reviewer #2: All comments have been addressed

Reviewer #3: All comments have been addressed

Reviewer #4: All comments have been addressed

publication criteria?

Reviewer #2: Yes

Reviewer #3: Partly

Reviewer #4: Yes

3. Has the statistical analysis been performed appropriately and rigorously?

Reviewer #2: Yes

Reviewer #3: Yes

Reviewer #4: Yes

4. Have the authors made all data underlying the findings in their manuscript fully available (please refer to the Data Availability Statement at the start of the manuscript PDF file)?

Reviewer #2: Yes

Reviewer #3: Yes

Reviewer #4: Yes

5. Is the manuscript presented in an intelligible fashion and written in standard English?

Reviewer #2: Yes

Reviewer #3: No

Reviewer #4: Yes

Reviewer #2: (No Response)

Reviewer #3: Thank you for addressing the earlier comments. Overall, the paper has improved significantly. However, the language quality is still subpar and requires extensive editing. Kindly address and infuse some new citations into the mix. Thank you.

Reviewer #4: I have some minor comments and clarifications which I have uploaded as a separate file.

**Do you want your identity to be public for this peer review?** For information about this choice, including consent withdrawal, please see our Privacy Policy

Reviewer #2: No

Reviewer #3: No

Reviewer #4: No

---

## [Editor Report · Decision Letter 5]

18 Feb 2026

PGPH-D-24-01717R5

Willingness and perceived ability to pay for Uganda’s proposed national health insurance scheme among informal sector workers in Iganga and Mayuge districts, Uganda: a contingent valuation method

Dear Dr. Noel Namuhani,

Thank you for submitting your manuscript to PLOS Global Public Health. After careful consideration, we feel that it has merit but does not fully meet PLOS Global Public Health’s publication criteria as it currently stands. Therefore, we invite you to submit a revised version of the manuscript that addresses the points raised during the review process.

We sincerely apologize for the delay in completing this review process. Upon reviewing the revised manuscript, it’s evident that the authors have addressed most of the comments raised. However, the response rate is still not clearly stated. While the sample size was calculated to be 853, it’s unclear if the authors achieved a 100% response rate. Were there any errors identified during data cleaning, or were there any observations dropped due to errors? It’s important to clarify whether all 853 individuals were invited and interviewed. This information should be included in the data analysis section of the manuscript.

Additionally, in the abstract, you mentioned that a total of 853 informal sector workers were randomly sampled. It’s crucial to specify if they were simply sampled or if they were both sampled and interviewed. Did all 853 participants take part in the survey? Please make this clear in your abstract.

Lastly, I noticed some words highlighted in different colors and font sizes throughout the manuscript. Please address these inconsistencies, specifically:

- The reference to "forms to the ODK server" found in the second paragraph of the data collection section.

- Table 5, which discusses multivariable factors under wealth quintiles.

Also, kindly ensure that all comments are removed.

Thank you

We look forward to receiving your revised manuscript.

Kind regards,

Ifunanya Clara Agu

Academic Editor
---

## [Editor Report · Decision Letter 6]

26 Feb 2026

Willingness and perceived ability to pay for Uganda’s proposed national health insurance scheme among informal sector workers in Iganga and Mayuge districts, Uganda: a contingent valuation method

PGPH-D-24-01717R6

Dear Mr Noel Namuhan,

We are pleased to inform you that your manuscript 'Willingness and perceived ability to pay for Uganda’s proposed national health insurance scheme among informal sector workers in Iganga and Mayuge districts, Uganda: a contingent valuation method' has been provisionally accepted for publication in PLOS Global Public Health.

Best regards,

Ifunanya Clara Agu

Academic Editor